# On the Absolute Stereochemistry of Tolterodine: A Circular Dichroism Study

**DOI:** 10.3390/ph12010021

**Published:** 2019-01-26

**Authors:** Marcin Górecki, Valerio Zullo, Anna Iuliano, Gennaro Pescitelli

**Affiliations:** 1Dipartimento di Chimica e Chimica Industriale, Università di Pisa, Via Moruzzi 13, 56124 Pisa, Italy; marcin.gorecki@icho.edu.pl (M.G.); valeriozullo@hotmail.it (V.Z.); anna.iuliano@unipi.it (A.I.); 2Institute of Organic Chemistry, Polish Academy of Sciences, Kasprzaka 44/52 St., 01-224 Warsaw, Poland

**Keywords:** absolute configuration, drug enantiomers, circular dichroism calculations, conformational analysis, variable-temperature CD, solvent dependent CD

## Abstract

Tolterodine (**1**) is a potent muscarinic receptor antagonist used in the treatment of overactive urinary bladder (OAB) syndrome. Tolterodine is chiral and it was patented, and is currently marketed, as the l-tartrate salt of the (*R*)-enantiomer. However, the existing literature does not offer an ultimate proof of a stereoselective mode of action of **1**. A second open stereochemical issue concerns the absolute configuration (AC) of **1**. Neither the original patents nor subsequent studies have established the AC of **1** in an unambiguous way, although the AC of the l-tartrate salt of **1** was assigned by X-ray diffractometry. Finally, neither electronic nor vibrational circular dichroism (ECD and VCD) spectra of **1** are reported so far. We performed a thorough ECD/VCD study of **1** in different solvents and at variable temperatures. Solvent and temperature dependence highlighted the existence of moderate flexibility which was confirmed by molecular modelling. ECD calculations with time-dependent density functional theory (TDDFT) accurately reproduced the experimental spectra and allowed us to confirm the AC of **1** in an independent way.

## 1. Introduction

Tolterodine (**1**, Figure 1) is a potent muscarinic receptor antagonist used in the treatment of overactive urinary bladder (OAB) syndrome, which causes incontinence and other symptoms [1,2,3]. Compared to other antimuscarinic drugs, it benefits from better tolerability and diminished side effects. Tolterodine (**1**) contains a single chirality center and may therefore exist as two different enantiomers. In general, the two enantiomers of a chiral substance show different pharmacology, ADME, and toxicology [4]. Tolterodine is marketed as the l-tartrate salt of the (*R*)-enantiomer (**2**), the form patented by Jönsson et al. in 1989 [5] and approved by FDA in 1998 [6]. Salt **2** is commonly named tolterodine tartrate [1], although a more correct name would be tolterodinium hydrogen tartrate. In the original patent, salt (+)-**2** was reported to have the same calcium antagonistic effect as the enantiomeric salt (−)-**2** and racemic tolterodine (**1**). The toxicities of the two enantiomers of salt **2** were also the same. Only for the anticholinergic activity, was (+)-**2** found to be 100 times more active than (−)-**2**. Very surprisingly, however, *rac*-**1** was as active as (+)-**2** as an anticholinergic. Reported IC_50_ values were 1.5 × 10^−2^ μM for *rac*-**1**, 1.3 × 10^−2^ μM for (+)-**2**, and 1.3 μM for (−)-**2**. These data do not offer sufficient proof of a stereoselective mode of action of tolterodine, although they still remain the only reported data about the (possible) drug stereoselectivity. To the best of our knowledge, in fact, the superiority of the (*R*)-enantiomer over the (*S*)- one for treating OAB syndrome has not been further proved. Almost all subsequent studies concerned the drug only in its (+)-**2** form [7,8]. Lately, (*S*)-tolterodine was also patented against urinary disorders [9]. When both enantiomers were tested for other types of activity, either no significant difference was found [10,11], or (*S*)-tolterodine displayed stronger activity [12,13]. Other antimuscarinic drugs like oxybutynin have an established enantioselective mode of action [14]. Because of the importance of tolterodine, it is not surprising that several enantioselective syntheses and enantiomer separations are reported to date, mainly to substantiate some new chiral auxiliary or catalyst [1]. According to our literature survey, however, obtaining enantiopure tolterodine may not be worth these efforts.

This is not the only open issue concerning the stereochemistry of tolterodine (**1**), the other one relates to its absolute configuration (AC). In the 1989 patent by Jönsson et al. [5], it was reported that the crystallization of *rac*-**1** with l-(+)-tartaric acid in ethanol afforded (+)-**2** salt with ${\left[\alpha \right]}_{546}^{25}$ + 36.0 (*c* unknown), whereas the same procedure with d-(−)-tartaric acid afforded (−)-**2** salt with ${\left[\alpha \right]}_{546}^{25}$ − 35.8. The reported elemental analysis clarifies that the obtained salts were hydrogen tartrates, i.e., with a 1:1 **1** to tartrate proportion as reported in Figure 1 for **2**. However, no correlation between the (+)/(−) sign of salt **2** and the AC of its component **1** was given. In a later patent by Gage and Cabaj [15], the same crystallization procedure of *rac*-**1** with l-(+)-tartaric acid in ethanol was repeated, affording (+)-**2** salt with ${\left[\alpha \right]}_{D}^{25}$ + 27.4 (*c* 1%, methanol) and a m.p. of 210–211 °C. This patent mentions for the first time the AC of **1**: the “racemic compound [*rac*-**1**] is later resolved in the conversion of tolterodine to (*R*)-tolterodine l-tartrate”. Very surprisingly, however, no details are given on how the (*R*)-configuration was established [15]. The two mentioned patents are often quoted as the proof of the (*R*)*-*configuration of component **1** in the (+)-**2** salt; from what said above, however, it is clear that such a claim is inappropriate. The first and only proof, to the best of our knowledge, of the AC of tolterodine was provided in 2005 by the X-ray determination of its salt **2** by Košutić-Hulita and Žegarac [16]. The species investigated was correctly named tolterodinium (2*R*,3*R*)-(+)-hydrogen tartrate (**2**); we note, however, that the “(2*R*,3*R*)-(+)” notation refers to the tartrate component of **2**, not to the salt itself. Using l-tartaric acid as internal reference, the AC of component **1** in the drug **2** was unambiguously identified as (*R*). Surprisingly enough, this paper has been quoted only five times to date, but never as *the* proof of the AC of tolterodine [1,17,18,19,20]. As a matter of fact, the correlation between the species investigated by X-ray and those described in the patents is unsure. Košutić-Hulita and Žegarac stated that the compound was prepared according to Gage and Cabaj’s procedure, however, 1,2-propylene carbonate was used as crystallization solvent instead of ethanol, the m.p. of the product was reported as 216 °C and, most importantly, the optical rotation (OR) of the salt **2** was not measured.

An unambiguous assignment of the AC of tolterodine (**1**) should obviously come from one of the several enantiospecific syntheses that have appeared in the literature [1]. The first enantiospecific synthesis, patented by Piccolo et al. in 2005 [21], used 6-methyl-4-phenylchroman-2-one (**3**) (Figure 1) as the stereodefinite intermediate. According to this patent, which is also often quoted as proof of the AC of tolterodine, (*S*)-(−)-**3** was converted into (*S*)-(−)-**1** by an enantiospecific route. The procedure was replicated by the same authors two years later in a paper, which also provided the first explicit correlation between the OR sign of **1** and **2** [22]. Chromanone **3** is a common intermediate in the route to **1** [23,24,25,26], although with unexpected implications. In fact, in a later report by Chen et al., the correlation was between (*R*)-(−)-**3** and (*R*)-(+)-**1** [23], that is, the AC of **3** was reversed with respect to Piccolo et al. [21,22]. A literature search on the reported OR values for 6-methyl-4-phenylchroman-2-one (**3**) confirms this inconsistency. Available OR data for (*R*)-**3** in chloroform are *negative* and span from ${\left[\alpha \right]}_{D}^{20}$ − 2.2 (*c* 0.3) to ${\left[\alpha \right]}_{D}^{20}$ − 6.2 (*c* 1) [23,26,27,28,29]. A very large *positive* value was however obtained in dichloromethane ${\left[\alpha \right]}_{D}^{20}$ + 36 (*c* 1) [24]. For (*S*)-**3**, Piccolo et al. reported a *negative*
${\left[\alpha \right]}_{D}^{20}$ − 2.8 (*c* 1.44, chloroform) [21,22]. In none of the syntheses of **1** involving **3** as intermediate [23,24,25,26], was the AC of this latter assigned independently from that of **1**. Since the AC of **1** was (incorrectly) given for established in the aforementioned patents [5,15,21], it is logical that the reported enantioselective syntheses of (*R*)-(+)-**1** were not further concerned with the stereochemical assignment [1].

From the above survey it appears that from the stereochemical point of view tolterodine (**1**) still has several aspects of interest. Additionally, we were surprised to find that no circular dichroism investigation of **1** has appeared to date. In 2018, Kirkpatrick et al. reported the electronic circular dichroism (ECD) data of tolterodine tartrate in a study about HPLC-ECD analysis of various drugs [30,31]. Very recently, some of us prepared tolterodine (+)-**1** with high enantiomeric purity (96%), starting from the optically active ethyl 3-phenyl-3-(2-hydroxy-5-methyl)phenylpropanoate (ee 96%) [32]. We took then the chance for a chiroptical investigation of **1** by means of electronic and vibrational CD (ECD and VCD), from both an experimental and computational point of view.

## 2. Results and Discussion

### 2.1. Sample Preparation

Tolterodine (+)-**1** was synthesized as described in Scheme 1. The enantioselective step was the Rh-catalyzed conjugate addition of phenylboronic acid to the ethyl 3-arylpropenoate **6**, in turn prepared in two steps from 5-methylsalicylaldehyde (**4**). This reaction, promoted by a deoxycholic-based binaphthyl phosphite, used as chiral ligand of Rh(I), gave optically active **7** with 96% ee. Hydrolysis of the ester group and reaction of the resulting carboxylic acid with diisopropyl amine in the presence of EDC, afforded, after chromatographic purification of the crude, the pure amide **9**. Reduction of the amide and hydrogenolysis of the benzyl protecting group gave (+)-tolterodine **1**, whose optical rotation was ${\left[\alpha \right]}_{D}^{20}$ + 24.9 (*c* 1.50, MeOH) [Lit. value: [22] ${\left[\alpha \right]}_{D}^{20}$ − 23.0 (*c* 1.5, MeOH) for (*S*)-tolterodine].

### 2.2. Experimental ECD Spectra

Tolterodine (**1**) contains two aromatic chromophores attached to the same carbon atom, which is also the only chirality center. The chromophores are a *p*-methylphenol and a phenyl chromophore [33,34]. ECD spectra of (+)-**1** were recorded in five different solvents, methylcyclohexane (MCH), chloroform, acetonitrile (ACN), methanol (MeOH) and water, and are shown in Figure 2.

They all display a weak and structured positive band between 260 and 300 nm (Δε = +1–2 M^−1^ cm^−1^), associated with a weak absorption in the same region (ε = 2000–3000 M^−1^ cm^−1^). These bands are allied with the ^1^L_b_ transitions of the substituted phenol and phenyl chromophores, occurring respectively at longer and shorter wavelength [35,36]. The presence of a long-wavelength tail in H_2_O lets us suspect that some aggregation occurs in this solvent even at the low concentration employed. The combination of ^1^L_a_ transitions of the two chromophores is responsible for the absorption and ECD bands in the 210–240 nm region [35,36]. All spectra feature a moderate positive ECD band around 235 nm (Δε = +4–8 M^−1^ cm^−1^) and a more intense negative band around 220 nm (Δε = −10–15 M^−1^ cm^−1^). The intensity and wavelength of both bands are affected by the solvent, but there is no regular trend following the solvent polarity. For the first positive band, the wavelength maximum decreases in the order CHCl_3_ > MCH > MeOH ~ H_2_O > ACN; for the second negative band, the wavelength order is MCH > H_2_O > ACN > MeOH (not recorded in CHCl_3_). For the first band, the intensity order is ACN > MeOH > MCH > H_2_O > CHCl_3_; for the second band, the intensity order is MCH > ACN > H_2_O > MeOH. At shorter wavelength, where ^1^B_b_ transitions occur, UV spectra are quite similar in all solvents but ECD spectra differ a lot. The ECD spectrum in MCH shows a strong positive band at 205 nm (Δε = +25.8 M^−1^ cm^−1^) with a short-wavelength shoulder, followed by a negative tail below 190 nm. The ECD spectrum in H_2_O shows a positive band at 207 nm (Δε = +5.6 M^−1^ cm^−1^) and three negative bands in the 190–200 region, the most intense of which is observed at 194 nm (Δε = −12.1 M^−1^ cm^−1^). The ECD spectrum in ACN is very weak around 205 nm but shows a moderately strong negative band at 199 nm (Δε = −10.7 M^−1^ cm^−1^) followed by a positive signal around 190 nm. Finally, the ECD spectrum in MeOH is weak below 210 nm. The relatively large variance of ECD spectra with the solvent, and the absence of a clear trend following the solvent polarity may only be justified with the occurrence of solvent-dependent conformational equilibria, favored by the molecular flexibility of **1**. It is useful to recall that ECD spectra depend not only on the absolute configuration, but also on the molecular conformation [37,38].

To further investigate the impact of the conformational freedom on the ECD spectra, variable temperature ECD (VT-ECD) spectra were recorded in MeOH and in MCH in the range from −80 to +20 °C (193–293 K), shown in Figure 3. The overall spectral shape is retained upon lowering the temperature in both solvents. The sign of the three major bands in the range 210–300 nm is preserved, accompanied by an intensity increase. For the spectra in MeOH, the increase is regular and more pronounced, and does not reach convergence at the lowest temperature used. The two stronger bands at 215 and 235 nm almost double their strength when passing from 20 °C to −80 °C (273 K to 193 K). In the same temperature range, the spectra in MCH experience a smaller intensity increase (25–30%) for both 220 and 235 nm bands. The trends observed in both solvents clearly indicate a fluxional situation at room temperature—especially in MeOH—which is quenched at low temperatures. A reason for the different behavior of the two solvents will be given below.

### 2.3. ECD Calculations and AC Assignment

To simulate the ECD spectrum of tolterodine (**1**), we used a well-established calculation procedure consisting of the following steps [39,40]: (1) conformational search with molecular mechanics; (2) geometry optimizations with density functional theory (DFT); (3) excited-state calculations on all relevant energy minima with time-dependent DFT (TDDFT); (4) Boltzmann averaging of calculated ECD spectra using DFT internal energies. 

A conformational search was run on (*R*)-**1** using a Monte Carlo algorithm and Merck Molecular Force Field (MMFF). Not surprisingly, the number of total conformers was huge: 247 within the pre-selected energy window of 10 kcal/mol, of which 54 within 5 kcal/mol, 4 within 2.5 kcal/mol, and only 2 within 2 kcal/mol. These latter showed an intramolecular O-H···N hydrogen bond defining an 8-membered cycle (see diagram in Figure 4). All 247 MMFF conformers were optimized with DFT at the ωB97X-D/6-31+G(d) level in vacuo, affording 89 optimized structures within 10 kcal/mol, 14 within 5 kcal/mol, and 4 within 2.5 kcal/mol (displayed in Figure 4). All the first 9 low-energy conformers showed the intramolecular O-H···N hydrogen bond. 

The first four low-energy conformers, covering 98.8% of the overall Boltzmann population, were considered for ECD simulations. Excited-state calculations were run with B3LYP and CAM-B3LYP functionals and def2-TZVP basis set in vacuo. Since ECD spectroscopy senses the chiral surrounding of the chromophores [38], it was anticipated that the reciprocal orientation of the aromatic rings would determine the ECD response. Such expectation was reinforced by our previous experience with chiral tetraarylmethane [41,42,43] and diarylmethane [44] derivatives. In the case of (*R*)-**1**, however, the intramolecular hydrogen bonding locks the two aromatic rings in a fixed position, as witnessed by the fact that the four low-energy conformers differ only in the conformation of the isopropyl groups. Not surprisingly then, the four calculated ECD spectra are very similar to each other (Figure 4). The resulting Boltzmann-weighted average spectra at 293 K and 193 K are compared in Figure 5 with the spectra measured in MCH at the same temperatures. It is expected that this solvent will not disturb the intramolecular hydrogen bonding, so that our low-energy DFT structures will represent correctly the conformational situation. As discussed above, ECD spectra in MCH are only slightly temperature-dependent, probably because the population of hydrogen-bonded structures is already large at room temperature and cannot be raised much by lowering the temperature. Moreover, solvent field effects on ECD spectra are very small for low-polarity solvents such as MCH [45], justifying our approach of running ECD calculations in vacuo. An attempt to reproduce the ECD spectra in H_2_O or MeOH would require an accurate treatment of hydrogen bonding which necessitates the inclusion of explicit solvent molecules [46,47,48,49], which is beyond the scope of our study. In fact, the result of VT-ECD experiments in MeOH suggests a more complicated conformational situation with respect to MCH, probably because of a temperature-dependent competition between solute-solvent hydrogen bonding and intramolecular hydrogen bonding.

The agreement between ECD spectra calculated in vacuo and experimental spectra measured in MCH is extremely good (Figure 5). The signs of the four bands above 200 nm are all replicated by the calculations. The wavelength spacing between the bands is also reproduced, with the only exception of ^1^L_b_ transitions whose energy is slightly overestimated. This is a well-known pitfall of TDDFT calculations of benzene derivatives [44,50,51,52,53]. Anyway, there are no doubts about the absolute configuration of the examined compound, which is certainly (*R*)-**1**. The similarity factor (SF) [54] between calculated and experimental spectra at 293 K is 0.92 and 0.98 for B3LYP and CAM-B3LYP calculations, respectively; for the wrong enantiomer (*S*)-**1**, SF is <0.01 for both functionals. The agreement is practically perfect (Figure 5) if we use for the comparison the spectrum measured in MCH at −80 °C (193 K), a temperature where only the first 2 DFT-optimized conformers retain an appreciable population. Thus, we confirm the absolute configuration (AC) of tolterodine to be (*R*)-(+)-**1**. This is the first assignment of the AC of tolterodine based on chiroptical spectroscopies. It is also independent of any chemical correlation with other chiral compounds, whose inconsistencies were mentioned in the Introduction.

### 2.4. Experimental and Calculated IR/VCD Spectra

The mid-IR and VCD spectra of (+)-**1** measured in CCl_4_ are reported in Figure 6. This is the best VCD spectrum we were able to obtain after several independent measurements on two different instruments. The VCD spectrum is not particularly weak, as there are several bands with a dissymmetry *g*-factor (Δε/ε) above 10^−4^. With the help of frequency calculations described below, one may recognize in the IR spectrum the aromatic C=C stretching vibrations around 1600 and 1500 cm^−1^, the C-H bending vibrations of the isopropyl groups at 1400 and 1370 cm^−1^, the methylene twisting and wagging modes at 1260 and 1250 cm^−1^, the C-N stretching vibrations at 1170 and 1150 cm^−1^, and the aromatic C-H in-plane bending vibrations around 1030 cm^−1^. Frequency calculations were first run on the 4 most stable conformers of (*R*)-**1** described above, after re-optimization at the B3LYP/6-311+G(d,p) level in vacuo. However, the Boltzmann-weighted average calculated VCD spectrum, constructed from DFT-computed internal energies, did not reproduce satisfactorily the experimental spectrum. To exclude the possibility that we missed some important conformers in the calculation, we repeated B3LYP/6-311+G(d,p) geometry optimizations and frequency calculations on the first 40 conformers obtained after the previous ωB97X-D/6-31+G(d) optimizations. Unfortunately, the conformer populations did not vary too much; the first 4 low-energy conformers still covered a large amount (89.8%) of the overall Boltzmann population. In fact, the weighted-average IR and VCD spectra did not differ appreciably from the first result. These final spectra are plotted in Figure 6. The agreement between experimental and calculated IR spectra is not bad, although some experimental IR bands are missing in the calculated spectrum (see asterisks in Figure 6). On the contrary, the agreement between experimental and calculated VCD spectra is poor. Although a few VCD bands are correctly predicted (correspondence highlighted by vertical bars in Figure 6), several moderately strong VCD bands are not reproduced at all by calculations (asterisked in Figure 6). 

Using the best-fitting parameters evaluated from IR spectral comparisons to plot the VCD spectra, we obtained a disappointingly low SF of 0.28 for the correct (*R*)-enantiomer, and a just smaller SF of 0.17 for the wrong (*S*)-enantiomer. The situation was not much better when the Boltzmann averages were constructed from DFT-computed free energies instead of internal energies, which led to a SF of 0.29 for the correct (*R*) enantiomer, and 0.14 for the wrong (*S*) enantiomer. It can be concluded that the AC of tolterodine (**1**) could not be assigned from our VCD study, while ECD results described above were fully satisfactory. The reason for this failure must be probably sought in the possible occurrence of solute aggregation in the chosen solvent (CCl_4_) at the relatively large concentration used for VCD measurements (0.18 M, 3 orders of magnitude larger than for ECD) [55]. It must be mentioned that, at the same concentration, tolterodine (**1**) was sparely soluble in CDCl_3_ and not soluble in CD_3_CN or D_2_O. The occurrence of aggregation is suggested by the already discussed presence of various experimental IR/VCD bands which have no calculated counterpart. A theoretical description of the effects of self-aggregation is possible but beyond the scope of the present manuscript [56,57].

## 3. Materials and Methods 

### 3.1. General Information

For experimental details on the preparation of (*R*)-(+)-tolterodine (**1**) and its characterization data, see [32]. ECD/UV spectra were recorded using a J-815 spectrometer (Jasco, Tokyo, Japan) at room temperature in spectroscopic grade solvents. Solutions with suitable concentrations (details are given in the legend of each figure) were measured in quartz cells with a path length of 0.1 cm. All spectra were recorded using a scanning speed of 100 nm min^−1^, a step size of 0.2 nm, a bandwidth of 1 nm, a response time of 0.5 s, and an accumulation of 4 scans. The spectra were background-corrected using the respective solvent spectra recorded under the same conditions. VT-ECD measurements were carried out by using an Optistat optical spectroscopy cryostat (Oxford Instruments, Abingdon, UK) attached to the sample chamber of ECD instrument, in the temperature range from +25 to −80 °C, using measurement parameters listed above. Baseline correction was done by subtracting the spectrum of a reference solvent obtained under the same conditions; all VT-ECD spectra were normalised using a concentration at 25 °C. VCD/IR spectra were collected using a Chiral*IR*-2X VCD spectrometer from BioTools Inc. (Jupiter, FL, USA) at a resolution of 4 cm^−1^ in the range of 2000–850 cm^−1^ in spectroscopic grade CCl_4_ for 6h. The spectrometer was equipped with dual sources and dual ZnSe photoelastic modulators (PEMs) optimized at 1400 cm^−1^. A solution was measured in a BaF_2_ cell with a path length of ~100 μm assembled in a rotating holder. Baseline correction was achieved by subtracting the spectrum of CCl_4_ recorded under the same conditions.

### 3.2. Computational Section

Conformational searches and preliminary DFT calculations were run with Spartan’16 (2017, Wavefunction, Irvine, CA, USA) using default grids and convergence criteria. DFT and TDDFT calculations were run with the Gaussian16 suite [58], using default grids and convergence criteria. A conformational search was run on (*R*)-**1** with molecular mechanics (Merck Molecular Force Field, MMFF) using the Monte Carlo algorithm implemented in Spartan’16, allowing all possible rotations around single bonds. All structures thus found were optimized with DFT at the ωB97X-D/6-31+G(d) level in vacuo in Spartan’16. Further optimizations and frequency calculations were run in Gaussian at B3LYP/6-311+G(d,p) level in vacuo. All relevant structures had zero imaginary frequencies. TDDFT calculations were run on ωB97X-D/6-31+G(d) optimized structures with both B3LYP and CAM-B3LYP functionals, and the def2-TZVP basis set. The calculations included 40 excited states (roots). Component spectra were averaged according to the Boltzmann distribution at 293 and 193 K using the populations estimated from internal and free energies. All spectra were averaged and plotted using the software SpecDis (v 1.71) [54,59] using the plotting parameters listed in the figure legends. SpecDis was also employed to estimate similarity factors (SF). 

## 4. Conclusions

The most important conclusion of our study is the assignment of the AC of tolterodine as (*R*)-(+)-**1**. This confirms the previous X-ray assignment of the hydrogen l-tartrate salt [16], which should be quoted as the only reliable reference for the AC of the title compound. On the contrary, the patents by Jönsson [5] and Gage and Cabaj [15] contain no proof of the claimed AC, while the patent by Piccolo et al. [21] reports a chemical correlation with 6-methyl-4-phenylchroman-2-one (**3**) whose AC is unsure. Several subsequent enantioselective syntheses are also based on the same intermediate, whose literature OR data are ambiguous. Our study offers the first assignment of the AC of tolterodine based on chiroptical spectroscopies, which is independent of any chemical correlation with other chiral compounds. Our assignment is based on the comparison between experimental ECD spectra, measured in different solvents and at variable temperatures, with TDDFT-calculated spectra, whose agreement was excellent. On the contrary, experimental VCD spectra were not reproduced by DFT calculations, possibly because of solute aggregation affecting experimental spectra.

It is also worth mentioning that our literature survey did not highlight the existence of a definite proof of the enantioselective mode of action of tolterodine. In the original patent, both calcium antagonistic and anticholinergic effects, as well as toxicity, were reported to be similar for the enantiomerically pure compound and the racemic one. Almost all subsequent studies concerned the drug only in its enantiomerically pure form, with no further evidence of its superiority over the racemate. Because of the importance of this drug, several enantioselective syntheses and enantiomer separations have been reported in years. However, further clinical data need be acquired to demonstrate the necessity of pursuing and using the drug in its enantiomerically pure form.

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
