# Peer review of "On the Absolute Stereochemistry of Tolterodine: A Circular Dichroism Study"

_pharmaceuticals, 2019, doi:10.3390/ph12010021_

Round 1
Reviewer 1 Report
This article by Pescitelli et al. is a circular dichroism study on tolterodine, an anticholinergic drug which is made chiral by the presence of one chirality centre.
The study is framed in a scenario where the literature data on the absolute configuration (AC) and the optical rotation of the chiral drug are controversial and/or somewhat ambiguous, and this is why the manuscript does deserve publication in Pharmaceuticals. However, since the authors faced a few problems with the VCD analysis (weak signal, poor agreement between experimental and calculated spectra, and occurrence of self-aggregation phenomena) with respect to the ECD study, I would suggest not including the VCD in the present paper.
Another concern is the solubility of tolterodine in water. Are the authors sure that the ECD spectra in water are significant?
Also, I would like to report the following minor items to be revised:
(1) Page 1, line 37: the sentence “The only reported stereoselective activity was the anticholinergic one” would benefit from a rephrasing for the sake of clarity.
(2) Page 2, Scheme 1: the picture should be renamed as “Figure 1” (it is not a scheme), and the next figures/schemes renamed accordingly.
(3) Page 2, line 54: the verb after “This” is missing.
(4) Page 2, line 56: please check the “l” of tartaric acid (it should be typed in small-caps font).
(5) Page 2, line 71: for the sake of clarity, replace the word “salt” with “drug”. The (2R,3R) configuration given in ref. [16], in fact, refers to the tartaric acid skeleton, and not to the amine.
(6) Page 4, line 120: remove the word “respectively” [pleonastic].
(7) Page 10, last paragraph (lines 316 to 324): the authors should limit their study to the chemical assignment of the AC rather than going deeply on the pharmacological activity of the drug. This last paragraph should be removed because the statements given on the activity of the two enantiomers (or the racemate) are not supported by clinical data.
(8) Please replace the term “question” with “issue” all over the manuscript.
Author Response
We thank the referee for his/her review.
Point-by-point response to referee’s comments:
1) Since the authors faced a few problems with the VCD analysis (weak signal, poor agreement between experimental and calculated spectra, and occurrence of self-aggregation phenomena) with respect to the ECD study, I would suggest not including the VCD in the present paper.
We think that presenting a case of non-applicability of VCD analysis is very instructive, especially because this technique is commonly recognized as more accurate for AC assignment than ECD, which is however not true in general. Moreover, referee 2 seems to disagree on this point, as he/she focused mainly on the VCD portion of the manuscript.
2) Another concern is the solubility of tolterodine in water. Are the authors sure that the ECD spectra in water are significant?
We are aware of limited solubility in water, as discussed in the paper. We think that the ECD spectra are significant because the most intense ECD bands are retained in comparison with the spectrum in methanol. There are some extra bands appearing only in water, for which we have commented in the text: “The presence of a long-wavelength tail in H2O lets us suspect that some aggregation occurs in this solvent even at the low concentration employed.”
3) (1) Page 1, line 37: the sentence “The only reported stereoselective activity was the anticholinergic one” would benefit from a rephrasing for the sake of clarity.
The sentence has been rephrased: “Only for the anticholinergic activity, (+)-2 was found 100 times more active than (–)-2.”
4) (2) Page 2, Scheme 1: the picture should be renamed as “Figure 1” (it is not a scheme), and the next figures/schemes renamed accordingly.
Done as requested.
5) (3) Page 2, line 54: the verb after “This” is missing.
Corrected as “This is not the only open question…”
6) (4) Page 2, line 56: please check the “l” of tartaric acid (it should be typed in small-caps font).
Corrected as suggested.
7) (5) Page 2, line 71: for the sake of clarity, replace the word “salt” with “drug”. The (2R,3R) configuration given in ref. [16], in fact, refers to the tartaric acid skeleton, and not to the amine.
Corrected as requested.
8) (6) Page 4, line 120: remove the word “respectively” [pleonastic].
Corrected as requested.
9) (7) Page 10, last paragraph (lines 316 to 324): the authors should limit their study to the chemical assignment of the AC rather than going deeply on the pharmacological activity of the drug. This last paragraph should be removed because the statements given on the activity of the two enantiomers (or the racemate) are not supported by clinical data.
We think that our literature survey, which has not been criticized or contradicted by either of the two reviewers, entirely supports the first part of the paragraph. We agree, however, that the last sentence “but such efforts may be unjustified, and it is very likely that racemic tolterodine might be as active as the enantiopure compound for the OAB syndrome treatment” was overstated and has been deleted. In its place, we concluded with the following more neutral statement: “However, further clinical data needs be acquired to demonstrate the necessity of pursuing and using the drug in its enantiomerically pure form.”
10) (8) Please replace the term “question” with “issue” all over the manuscript.
Corrected as suggested (two occurrences).
Reviewer 2 Report
*English needs editing including ‘chirality centre’ should read ‘chiral centre’
*A Figure with the tolterodine molecular structure would help the reader.
*The MCH spectra in Figures 1 and 2 look different to me. Figure 2 look a mess. If temperature dependence is to be included some higher temperatures would be helpful.
*I cannot assess the quality of the calculations
*Figure 4a looks like a very poor agreement to me. I am happier with the -80 degree overlay. If it is so good then the argument about band position being wrong and OK for Figure 4 (left) is suspicious to me. The assignment based on the lower wavelength bands is OK.
*The VCD is very disappointing. The authors don’t show the absorbance magnitude of their spectra – were they too high? How was the path length measured?
*What does * mean in Figure 5? Solvent? Assuming that is the case, and being most optimistic about the calculation validity (reasonable given vibrations are easier than electronic calculations), then it very much looks to me as if the VCD has a non-flat baseline. I strongly recommend the authors look at their baseline correction. I think this will improve the VCD significantly. They should also show total absorbance (it needs to be less than 1).
Author Response
We thank the referee for his/her review.
Point-by-point response to referee’s comments:
1) English needs editing including ‘chirality centre’ should read ‘chiral centre’
We checked once again English language throughout the manuscript. Concerning the stereochemical nomenclature: the IUPAC Gold Book lists the term “chirality centre” or “centre of chirality” as “an atom holding a set of ligands in a spatial arrangement which is not superposable on its mirror image”, while the single term chiral is much less specific: “having the property of chirality”. According to Ernest Eliel, “ [chiral] adjectives are often used loosely in English, for example ‘‘chiral center’’ is used properly for ‘‘center of chirality.’” (Chirality, 9:428–430 (1997)). Therefore, the nomenclature “centre of chirality” is perfectly appropriate and even preferable to “chiral centre”.
2) A Figure with the tolterodine molecular structure would help the reader.
Figure 1 and Scheme 1 show the molecular diagram of tolterodine, while Figure 4 shows the 3D molecular structure. It is not clear to us if the referee is requesting anything different.
3) The MCH spectra in Figures 1 and 2 look different to me. Figure 2 look a mess. If temperature dependence is to be included some higher temperatures would be helpful.
MCH spectra at 20°C are pretty the same in the two figures; the only different is in the X-axis limits, which span 190-320 nm in Figure 1 (now Figure 2) and 200-320 nm in Figure 2 (now Figure 3). The minor differences are due to the VT apparatus which fits between the sample holder and the photomultiplier, changing the effective optical path. We cannot agree that “Figure 2 look a mess”: the CD spectra evolve with temperature in a rather regular way; we used different blue hues to emphasize the progressive cooling. Low temperatures were investigated, and not higher ones, because the aim of the VT experiments was to quench conformational equilibria.
4) I cannot assess the quality of the calculations
The calculation procedure we employed is well established and documented by several applications; please see refs. [39,40].
5) Figure 4a looks like a very poor agreement to me. I am happier with the -80 degree overlay. If it is so good then the argument about band position being wrong and OK for Figure 4 (left) is suspicious to me. The assignment based on the lower wavelength bands is OK.
In Figure 4a, the agreement between the experimental spectrum and calculated ones is acceptable with B3LYP functional and very good with CAM-B3LYP. This is witnessed by the reported similarity factors: “The similarity factor (SF) [54] between calculated and experimental spectra at 293K is 0.92 and 0.98 for B3LYP and CAM-B3LYP calculations, respectively; for the wrong enantiomer (S)-1, SF is<0.01 for both functionals.” Honestly speaking, we don’t fully understand referee’s sentence “the argument about band position being wrong and OK is suspicious to me”. The only band with “wrong” position is the 1Lb, as commented in the text.
6) The VCD is very disappointing. The authors don’t show the absorbance magnitude of their spectra – were they too high? How was the path length measured?
We agree that the quality of the VCD spectrum is low. As stated in the manuscript, we did our best to record a meaningful spectrum by replicating the measurements on two different instruments (from Jasco and BioTools). The reported spectrum is the best we could obtain in terms of S/N ratio. The cell path length is declared by the vendor and was checked by the usual interferometric procedure based on fringing effect. In Figure 6 we have added on the right Y axis the measured absorbance. All peaks are well below 1 u.a.
7) What does * mean in Figure 5? Solvent? Assuming that is the case, and being most optimistic about the calculation validity (reasonable given vibrations are easier than electronic calculations), then it very much looks to me as if the VCD has a non-flat baseline. I strongly recommend the authors look at their baseline correction. I think this will improve the VCD significantly. They should also show total absorbance (it needs to be less than 1).
The asterisked peaks are those for which no calculated counterpart was found; this is now additionally stated in the figure caption. As suggested by the referee, we run again our baseline correction by subtracting the spectrum recorded for the solvent in the same cell and measurement conditions and got an improved spectrum which is presented in Figure 6. Although the baseline looks flatter than before, the agreement between calculated and experimental spectrum was not much increased. However, the difference of similarity factor between the correct and wrong enantiomer is now slightly larger. In Figure 6 we have added on the right Y axis the measured absorbance. All peaks are well below 1 u.a.
In summary, this latter comment led to the following changes:
(a) a new Figure 6 is provided with: improved experimental VCD spectrum (flatter baseline); vertical bars highlighting the correspondence between experimental and calculated bands; a second Y axis with absorbance values;
(b) in the caption of Figure 6, we added the sentence: “The vertical bands highlight the correspondence between experimental and calculated VCD peaks, while the asterisks (*) indicate experimental IR/VCD peaks with no calculated counterpart.”
(c) in the text, lines 252-253, we added the sentence “Although a few VCD bands are correctly predicted (correspondence highlighted by vertical bars in Figure 6)”
(d) the SF values were recalculated and corrected according to the new experimental spectrum.
Round 2
Reviewer 1 Report
I think that now the paper can be published.
Author Response
No action requested.
Reviewer 2 Report
My request for a structure was intended to be for one with the 3D relative geometries indicated.
Author Response
We added a molecular diagram in Figure 4 to illustrate the 8-membered hydorgen bonded cycle seen in low-energy minima.